# Safe Reinforcement Learning
# via Curriculum Induction

**Matteo Turchetta**[*]
Department of Computer Science
ETH Zurich
matteotu@inf.ethz.ch

**Andrey Kolobov**
Microsoft Research
Redmond, WA-98052
akolobov@microsoft.com

**Shital Shah**
Microsoft Research
Redmond, WA-98052
shitals@microsoft.com

**Andreas Krause**
Department of Computer Science
ETH Zurich
krausea@ethz.ch

**Alekh Agarwal**
Microsoft Research
Redmond, WA-98052
alekha@microsoft.com

## Abstract

In safety-critical applications, autonomous agents may need to learn in an environment where mistakes can be very costly. In such settings, the agent needs to behave safely not only *after* but also *while* learning. To achieve this, existing safe reinforcement learning methods make an agent rely on priors that let it avoid dangerous situations during exploration with high probability, but both the probabilistic guarantees and the smoothness assumptions inherent in the priors are not viable in many scenarios of interest such as autonomous driving. This paper presents an alternative approach inspired by human teaching, where an agent learns under the supervision of an automatic instructor that saves the agent from violating constraints during learning. In this new model, the instructor needs to know neither how to do well at the task the agent is learning, nor how the environment works. Instead, it has a library of reset controllers that it activates when the agent starts behaving dangerously, preventing it from doing damage. Crucially, the choices of which reset controller to apply in which situation affect the speed of agent learning. Based on observing agents' progress, the teacher itself learns a policy for choosing the reset controllers, *a curriculum*, to optimize the agent's final policy reward. Our experiments use this framework in two challenging environments to induce curricula for safe and efficient learning.

## 1 Introduction

Safety is a major concern that prevents application of reinforcement learning (RL) [45] to many practical problems [16]. Among the RL safety notions studied in the literature [23], ensuring that the agent does not violate constraints is perhaps the most important. Consider, for exmple, training a policy for a self-driving car's autopilot. Although simulations are helpful, much of the training needs to be done via a process akin to RL on a physical car [26]. At that stage, it is critical to avoid damage to property, people, and the car itself. Safe RL techniques aim to achieve this primarily by imparting the agent with priors about the environment and equipping it with sound ways of updating this information with observations [17, 9, 12, 13]. Some do this heuristically [2], while others provide safety guarantees through demonstrations [47], or by assuming access to fairly accurate dynamical models [4], or at the cost of smoothness assumptions [9, 28, 48, 49]. These assumptions hold, e.g., in certain drone control scenarios [8] but are violated in settings such as autonomous driving, where a small delta in control inputs can make a difference between safe passage and collision.

---

[*]The author did part of this work while at Microsoft Research, Redmond.

In this paper, we propose *C*urriculum *I*nduction for *S*afe *R*einforcement learning (CISR, *"Caesar"*), a safe RL approach that lifts several prohibitive assumptions of existing ones. CISR is motivated by the fact that, as humans, we successfully overcome challenges similar to those in the autopilot training scenario when we help our children learn safely. Children possess inaccurate notions of danger, have difficulty imitating us at tasks requiring coordination, and often ignore or misunderstand requests to be careful. Instead, e.g., when they learn how to ride a bicycle, we help them do it safely by first equipping the bicycle with training wheels, then following the child while staying prepared to catch them if they fall, finally letting them ride freely but with elbow and knee guards for some time, and only then allowing them to ride like grown-ups. Importantly, each "graduation" to the next stage happens based on the learner's observed performance under the current safeguard mechanism.

**Key ideas.** In CISR, an artificial teacher helps an agent (student) learn potentially dangerous skills by inducing a sequence of safety-ensuring training stages called *curriculum*. A student is an RL agent trying to learn a policy for a constrained MDP (CMDP) [6]. A teacher has a decision rule – a *curriculum policy* – for constructing a curriculum for a student given observations of the student's behavior. Each curriculum stage lasts for some number of RL steps of the student and is characterized by an *intervention* (e.g., the use of training wheels) that the teacher commits to use throughout that stage. Whenever the student runs the risk of violating a constraint (falling off the bike), that stage's intervention automatically puts the agent into a safe state (e.g., the way training wheels keep the bike upright), in effect by temporarily overriding the dynamics of the student's CMDP. The teacher's curriculum policy chooses interventions from a pre-specified set such as {*use of training wheels, catching the child if they fall, wearing elbow and knee guards*} with the crucial property that any single intervention from this set keeps the agent safe as described above. A curriculum policy that commits to any one of these interventions for the entire learning process is sufficient for safety, but note that in the biking scenario we don't keep the training wheels on the bike forever: at some point they start hampering the child's progress. Thus, the teacher's natural goal is to *optimize the curriculum policy* with respect to the student's policy performance at the end of the learning process, assuming the process is long enough that the student's rate of attempted constraint violations becomes very small. In CISR, the teacher does this via a round-based process, by playing a curriculum policy in every round, observing a student learn under the induced curriculum, evaluating its performance, and trying an improved curriculum policy on a new student in the next round.

**Related Work.** CISR is a form of *curriculum learning (CL)* [39]. CL and *learning from demonstration (LfD)* [15] are two classes of approaches that rely on a teacher as an aid in training a decision-making agent, but CISR differs from both. In LfD, a teacher provides demonstrations of a good policy for the task at hand, and the student uses them to learn its own policy by behavior cloning [37], online imitation [35], or apprenticeship learning [1]. In contrast, CISR does not assume that the teacher has a policy for the student's task at all: e.g., a teacher doesn't need to know how to ride a bike in order to help a child learn to do it. CL generally relies on a teacher to structure the learning process. A range of works [33, 22, 21, 41, 52, 38, 51] explore ways of building a curriculum by modifying the learning environment. CISR is closer to Graves et al. [24], which uses a fixed set of environments for the student and also uses a bandit algorithm for the teacher, and to [34], which studies how to make the teacher's learning problem tractable. CISR's major differences from existing CL work is that (1) it is the first approach, to our knowledge, that uses CL for *ensuring safety* and (2) uses *multiple* students for training the teacher, which allows it to induce curricula in a more data-driven, as opposed to heuristic, way. Regarding safe RL, in addition to the literature mentioned above, Le et al. [30], which proposes a CMDP solver, considers the same training and test safety constraints as ours. In that work, the student avoids potentially unsafe environment interaction by learning from batch data, which places strong assumptions on MDP dynamics and data collection policy neither verifiable nor easily satisfied in practice [42, 11, 3]. We use the same solver, but in an online setting.

The ideas introduced in this work may be applicable in several kinds of safety-sensitive settings where CISR can be viewed as a *meta-learning framework* [50], with curriculum policy as a "hyperparameter" being optimized. In our experiments, the number of iterations CISR needs to learn a good curriculum policy is small. This allows its use in robotics, where a curriculum policy is trained on agents with one set of sensors and applied to training agents with different sensors of similar capabilities, e.g., as in Pan et al. [36] for autonomous rovers. Further promising scenarios are training a curriculum policy in simulation and applying it to physical agents and using CISR in *intelligent tutoring systems* [14].

**Contributions.** Our main contributions are: (1) We introduce CISR, a novel framework for exploiting prior knowledge to guarantee safe training and deployment in RL that forgoes many unrealistic assumptions made in the existing safe RL literature. (2) We present a principled way of optimizing

curriculum policies across generations of students while guaranteeing safe student training. (3) We show empirically in two environments that students trained under CISR-optimized curricula attain reward performance comparable or superior to those trained without a curriculum and remain safe throughout training, while those trained without a curriculum don't. (4) We release an open source implementation of CISR and of our experiments[2].

## 2 Background: Constrained Markov Decision Processes

In this work, we view a learning agent, which we will call a *student*, as performing constrained RL. This framework has been strongly advocated as a promising path to RL safety [40], and expresses safety requirements in terms of an *a priori unknown* set of feasible safe policies that the student should optimize over. In practice, this feasible policy set is often described by a *constrained Markov decision process (CMDP)* [6]. We consider CMDPs of the form $\mathcal{M} = \langle \mathcal{S}, \mathcal{A}, \mathcal{P}, r, \mathcal{D} \rangle$, where $\mathcal{S}$ and $\mathcal{A}$ are a state and action space, respectively, $\mathcal{P}(s'|s, a)$ is a transition kernel, $r : \mathcal{S} \times \mathcal{A} \times \mathcal{S} \to \mathbb{R}$ is a reward function, and $\mathcal{D}$ is a set of unsafe terminal states. We focus on settings where safety corresponds to avoiding visits to the set $\mathcal{D}$. The student's objective, then, is to find a policy $\pi : \mathcal{S} \to \Delta_{\mathcal{A}}$, i.e., a mapping from states to action distributions, that solves the following constrained optimization problem, where $\rho^{\pi}$ is a distribution of trajectories induced by $\pi$ and $\mathcal{P}$ given some fixed initial state distribution:

$$\pi^* = \arg\max_{\pi} \ \mathbb{E}_{\rho^\pi} \sum_{t=0}^{T} r(s_t, a_t, s_{t+1}), \quad \text{s.t.} \quad \mathbb{E}_{\rho^\pi} \sum_{t=0}^{T} \mathbb{I}(s_t \in \mathcal{D}) \leq \kappa, \tag{1}$$

where $\mathbb{I}$ is the indicator function. To ensure complete safety, we restrict our attention to problems where the value of $\kappa$ makes the constraints feasible. While we have presented the finite-horizon undiscounted version of the problem, both the objective and the constraint can be expressed as the average or a discounted sum over an infinite horizon. For a generic CMDP $\mathcal{M}$ we denote the set of its feasible policies as $\Pi_{\mathcal{M}}$, and the value of any $\pi \in \Pi_{\mathcal{M}}$ as $V_{\mathcal{M}}(\pi)$.

There are a number of works on solving CMDPs that find a nearly feasible and optimal policy with sufficiently many trajectories, but violate constraints during training [2, 13]. In contrast, we aim to enable the student to learn a policy for CMDP $\mathcal{M}$ *without violating any constraints in the process*.

## 3 Curriculum Induction for Safe RL

We now describe CISR, our framework for enabling a student to learn without violating safety constraints. To address the seemingly impossible problem of students learning safely in an unknown environment, CISR includes a *teacher*, which is a learning agent itself. The teacher serves two purposes: (1) mediating between the student and its CMDP in order to keep the student safe, and (2) learning a mediation strategy – a curriculum policy – that helps the student learn faster. First we formally describe the teacher's mediation tools called *interventions* (Section 3.1). Then, in Section 3.2 we show how, from the student's perspective, each intervention corresponds to a special CMDP where training is safe and every feasible policy is also feasible in original CMDP, ensuring that objective (1) is always met. Finally, in Section 3.3 we consider the teacher's perspective and show how, in order to optimize objective (2), it iteratively improves its curriculum policy by trying it out on different students.

### 3.1 Interventions

In CISR, the teacher has a set of *interventions* $\mathcal{I} = \{\langle \mathcal{D}_i, \mathcal{T}_i \rangle\}_{i=1}^{K}$. Hereby, each intervention is defined by a set $\mathcal{D}_i \subset \mathcal{S}$ of *trigger states* where this intervention applies and $\mathcal{T}_i : \mathcal{S} \to \Delta_{\mathcal{S} \setminus \mathcal{D}_i}$, a state-conditional reset distribution. The semantics of an intervention is as follows. From the teacher's perspective, $\mathcal{D}_i$ is a set of undesirable states, either because $\mathcal{D}_i$ intersects with the student CMDP's unsafe state set $\mathcal{D}$ or because the student's current policy may easily lead from $\mathcal{D}_i$'s states to $\mathcal{D}$'s. Whenever the student enters a state $s \in \mathcal{D}_i$, the teacher can *intervene* by resetting the student to another, safe state according to distribution $\mathcal{T}_i(\cdot|s)$. We assume the following about the interventions:

**Assumption 1** (**Intervention set**). *a) The intervention set $\mathcal{I}$ is given to the teacher as input and is fixed throughout learning. b) The interventions in $\mathcal{I}$ cannot be applied after student learning.*

Assumption a) is realistic in many settings, where the student is kept safe by heuristics in the form of simple controllers such as those that prevent drones from stalling. At least one prior work, Eysenbach

et al. [18], focuses on how safety controllers can be learned, although their safety notion (policy reversibility) is much more specialized than in CISR. Assumption b) is realistic in that a safety controller practical enough to be used beyond training is likely to be part of the agent as in [5], removing the need for safety precautions during training. An example of a safety mechanism that cannot be used beyond training is motion capture lab equipment to prevent collisions.

### 3.2 The student's problem

We now describe the student's learning process under single and multiple interventions of the teacher. As we explain here, training in the presence of an interventions-based teacher can be viewed as learning in a *sequence* of CMDPs that guarantee student safety under simple conditions. Later, in Sec. 3.3, we formalize these CMDP sequences as *curricula*, and show how the teacher can induce them using a *curriculum policy* in order to accelerate students' learning progress.

**Intervention-induced CMDPs.** Fix an intervention $\langle \mathcal{D}_i, \mathcal{T}_i \rangle$ and suppose the teacher commits to using it throughout student learning. As long as the student avoids states in $\mathcal{D}_i$, deemed by the teacher too dangerous for the student's ability, the student's environment works like the original CMDP, $\mathcal{M}$. But whenever the student enters an $s \in \mathcal{D}_i$, the teacher leads it to a safe state $s' \sim \mathcal{T}_i(\cdot|s), s' \notin \mathcal{D}_i$.

Thus, each of teacher's interventions $i \in \mathcal{I}$ induces a student CMDP $\mathcal{M}_i = \langle \mathcal{S}, \mathcal{A}, \mathcal{P}_i, r_i, \mathcal{D}, \mathcal{D}_i \rangle$, where $\mathcal{S}$ and $\mathcal{A}$ are as in the original CMDP $\mathcal{M}$, but the dynamics are different: for all $a \in \mathcal{A}$, $\mathcal{P}_i(s'|s, a) = \mathcal{P}(s'|s, a)$ for all $s \in \mathcal{S} \backslash \mathcal{D}_i$ and $\mathcal{P}_i(s'|s, a) = \mathcal{T}_i(s'|s)$ for $s \in \mathcal{D}_i$. The reward function is modified to assign $r_i(s, a, s') = 0$ for $s \in \mathcal{D}_i, s' \notin \mathcal{D}_i$: all student's actions in these cases get overridden by the teacher's intervention, having no direct cost for the student. However, the teacher cannot supervise the student forever; the student must learn a safe and high-return policy that does not rely on its help. We thus introduce a constraint on the number of times the student can use the teacher's help. This yields the following problem formulation for the student, where $\rho_i^\pi$ is a distribution of trajectories induced by $\pi$ given some fixed initial state distribution and the modified transition function $\mathcal{P}_i$:

$$\pi^* = \arg\max \mathbb{E}_{\rho_i^\pi} \sum_{t=0}^{T} r_i(s_t, a_t, s'_{t+1}), \text{ s.t. } \mathbb{E}_{\rho_i^\pi} \sum_{t=0}^{T} \mathbb{I}(s_t \in \mathcal{D}) \leq \kappa_i, \mathbb{E}_{\rho_i^\pi} \sum_{t=0}^{T} \mathbb{I}(s_t \in \mathcal{D}_i) \leq \tau_i, \quad (2)$$

where $\kappa_i \geq 0$ and $\tau_i \geq 0$ are intervention-specific tolerances set by the teacher. Thus, although the student doesn't incur any cost for the teacher's interventions, they are associated with violations of teacher-imposed constraints. Our CMDP solver [30], discussed in Section 4, penalizes the student for them during learning, making sure that the student doesn't exploit them in its final policy.

By construction, each intervention-induced CMDP $\mathcal{M}_i$ has two important properties, which we state below and prove in Appendix C. First, if the teacher has lower tolerance for constraint violations than the original CMDP, an optimal learner operating in $\mathcal{M}_i$ will eventually come up with a policy that is safe in its original environment $\mathcal{M}$:

**Proposition 1** (*Eventual safety*). *Let $\Pi_{\mathcal{M}}$ and $\Pi_{\mathcal{M}_i}$ be the sets of feasible policies for the problems in Equations* (1) *and* (2)*, respectively. Then, if $\tau_i + \kappa_i \leq \kappa$, $\Pi_{\mathcal{M}_i} \subseteq \Pi_{\mathcal{M}}$.*

Intuitively, once the teacher is removed, the student can fail either by passing through states where it used to be rescued or by reaching states where the teacher was not able to save it in the first place; $\tau_i + \kappa_i \leq \kappa$ ensures that the probability of either of these cases is sufficiently low that a feasible policy in $\mathcal{M}_i$ is also feasible in $\mathcal{M}$. While this guarantees the student's *eventual* safety, it doesn't say anything about *safety during learning*. The second proposition states conditions for learning safely:

**Proposition 2** (*Learning safety*). *Let $\mathcal{D}$ be the set of unsafe states of CMDPs $\mathcal{M}$ and $\mathcal{M}_i$, and let $\mathcal{D}_i$ be the set of trigger states of intervention $i$. If $\mathcal{D} \subseteq \mathcal{D}_i$ and $\mathcal{P}(s'|a, s) = 0$ for every $s' \in \mathcal{D}$, $s \in \mathcal{S} \setminus \mathcal{D}_i$, and $a \in \mathcal{A}$, then an optimal student learning in CMDP $\mathcal{M}_i$ will not violate any of $\mathcal{M}$'s constraints throughout learning.*

Informally, Proposition 2 says that if the set of trigger states $\mathcal{D}_i$ of the teacher's intervention "blankets" the set of unsafe states $\mathcal{D}$, the student has no way of reaching states in $\mathcal{D}$ without triggering the intervention and being rescued first, and hence is safe even when it violates $\mathcal{M}_i$'s constraints.

**Assumption 2** (*Intervention safety*). *In the rest of the paper, we assume all teacher interventions to meet the conditions of Proposition 2.*

We make this assumption for conceptual simplicity, but it can hold in reality: systems such as aircraft stall prevention and collision avoidance guarantee near-absolute safety. Even in the absence thereof,

CISR informally keeps the student as safe during training as teacher's interventions allow. In Sec. 5, we show that, even when this assumption is violated and the interventions cannot guarantee absolute safety, CISR still improves training safety by three orders of magnitude over existing approaches.

**Sequences of intervention-induced CMDPs and knowledge transfer.** As suggested by the biking example, the student's learning is likely to be faster under a *sequence* of teacher interventions, resulting in a sequence of CMDPs $\mathcal{M}_{i_1}, \mathcal{M}_{i_2}, \ldots$. This requires a mechanism for the student to carry over previously acquired skills from one CMDP to the next. We believe that most knowledge transfer approaches for unconstrained MDPs, such as transferring samples [29], policies [20], models [19] and values [46], can be applied to CMDPs as well, with the caveat that the transfer mechanism should be tailored to the environment, teacher's intervention set, and the learning algorithm the student uses. In Section 4, we present the knowledge transfer mechanism used in our implementation.

### 3.3 The teacher's problem

Given that, under simple conditions, any sequence of teacher's interventions will keep the student safe, the teacher's task is to sequence interventions/CMDPs for the student so that the student learns the highest-expected-reward policy. In CISR, the teacher does this by iteratively trying a curriculum policy on different students and improving it after each attempt. This resembles human societies, where curriculum policies are implicitly learned through educating generations of students. At the same time, it is different from prior approaches such as Graves et al. [24], Matiisen et al. [31], which try to learn and apply a curriculum on the same student. These approaches embed a fixed, heuristic curriculum policy within the teacher to induce a curriculum but cannot improve this policy over time. In contrast, CISR exploits information from previous students to optimize its curriculum policy following a data-driven approach.

---

**Algorithm 1** CISR

1: **Input**: Interventions $\mathcal{I}$, Initial teacher $\pi_0^T$
2: **for** $j = 0, 1, \ldots, N_t$ **do**
3:     $\pi_{0,j} \leftarrow$ get_student()
4:     **for** $n = 0, 1, \ldots, N_s$ **do**
5:         $\mathcal{M}_{i_n} \leftarrow \pi_j^T(o_0^T, \ldots, o_n^T)$
6:         **if** $n > 0$ **then** $\pi_{n,j} \leftarrow$ transfer$(\pi_{n-1,j})$
7:         $\pi_{n,j} \leftarrow$ student.train$(\mathcal{M}_{i_n})$
8:         $o_n^T \leftarrow \phi(\pi_{n,j})$
9:     $\pi_{j+1}^T \leftarrow$ teacher.train$(\{(\pi_k^T, \hat{V}(\pi_{N_s,k}))\}_{k=1}^j)$

---

**What does *not* need to be assumed of the teacher.** CISR makes very few assumptions about the teacher's abilities. In particular, while the teacher needs to be able to evaluate the student's progress, it can do so according to its internal notion of performance, which may differ from the student's one. Thus, the teacher does not need to know the student's reward. Moreover, since the interventions are given, the teacher does not need to know the student's dynamics. However, the intervention design may still require *approximate* and *local* knowledge of the dynamics, which is much less restrictive than *perfect* and *global* knowledge. Furthermore, the teacher does not need to have a policy for performing the task that the student is trying to learn nor to be able to communicate the set $\mathcal{D}_i$ of an intervention's trigger states for any $i$ to the student. It only needs to be able to *execute* any intervention in $\mathcal{I}$ without violating laws governing the CMPD $\mathcal{M}$, i.e., using conditional reset distributions that are realizable according to $\mathcal{M}$'s dynamics. However, the teacher is not assumed to use only the student's action and observation set $\mathcal{A}$ and $\mathcal{S}$ to execute the interventions — it may be able to do things the student can't, such as setting the the student upright if it is falling from a bike.

**In CISR the teacher learns online.** Abstractly, we view the teacher as an online learner presented in Algorithm 1. In particular, for rounds $j = 1, \ldots, N_t$:

1. The teacher plays a *decision rule* $\pi_j^T$ that makes a *new* student $j$ learn under an adaptively constructed sequence $C_j = (\mathcal{M}_{i_1}, \ldots, \mathcal{M}_{i_{N_s}})$ of intervention-induced CMDPs (lines 4-8).

2. Each student $j$ learns via a total of $N_s$ interaction units (e.g., steps, episodes, etc.) with an environment. During each unit, it acts in a CMDP in $C_j$. It updates its policy by transferring knowledge across interaction units (Line 6). The teacher computes *features* $\phi(\pi_{n,j})$ of student $j$'s performance (lines 8) by evaluating $j$'s policies throughout $j$'s learning process. Based on them, the teacher's decision rule proposes the next intervention MDP in $C_j$.

3. The teacher adjusts its decision rule's parameters (line 9) that govern how a CMDP sequence $C_{j+1}$ will be produced in the next round.

**Assumption 3** (*Length of student learning*). *For all potential students, their CMDP solvers* `student.train` *are sound and complete.*[3] $N_s$ *is much larger than the number of interactions it takes the solver to find a feasible policy* $\pi \in \Pi_{\mathcal{M}_i}$ *for at least one intervention CMDP* $\mathcal{M}_i$, $i \in \mathcal{I}$.

This assumption ensures that students can, in principle, learn a safe policy in the allocated amount of training $N_s$ under some intervention sequence. It also allows the teacher to learn to induce such sequences, given enough rounds $N_t$, even though not every student trained in the process will necessarily have a feasible policy for $\mathcal{M}$ at the end of its learning.

This framework's concrete instantiations depend on the specifics of (i) the decision rule that produces a sequence $C_j$ in each round, (ii) the teacher's evaluation of the student to estimate $\hat{V}(\pi_{N_s,k})$ in each round. Next, we consider each of these aspects.

**Curricula and curriculum policies (i).** Before discussing teacher's decision rules that induce intervention sequences in each round, we formalize the notion of these sequences themselves:

**Definition 1** (*Curriculum*). *Suppose a student learns via* $N_s$ *interaction units (e.g., steps or episodes) with an environment, and let* $\mathcal{I}$ *be a set of teacher's interventions. A curriculum* $C$ *is a sequence* $\mathcal{M}_{i_1}, \ldots, \mathcal{M}_{i_{N_s}}$ *of length* $N_s$ *of CMDPs s.t. the student interacts with CMDP* $\mathcal{M}_{i_n}$ *during unit* $n$, *where* $\mathcal{M}_{i_n}$ *is induced by an intervention* $i_n \in \mathcal{I}$.

The difference between a curriculum and a teacher's decision rule that produces it is crucial. While a decision rule for round $j$ *can* be a mapping $C_j : [N_s] \rightarrow \mathcal{I}$ exactly like a curriculum, in general it is useful to make it depend on the student's policy $\pi_{n,j}$ at the start of each interaction unit $n$. In practice, the teacher doesn't have access to $\pi_{n,j}$ directly, but can gather some statistics $\phi(\pi_{n,j})$ about it by conducting an evaluation procedure that we discuss shortly. Examples of useful statistics include the number of times the student's policy triggers teacher's interventions, features of states where this happens, and, importantly, an estimate of the policy value $\hat{V}(\pi_{n,j})$.

Thus, an adaptive teacher is an agent operating in a *partially observable* MDP $\langle \mathcal{S}^T, \mathcal{A}^T, \mathcal{P}^T, \mathcal{R}^T, \mathcal{O}^T, \mathcal{Z}^T \rangle$, where $\mathcal{S}^T = \overline{\Pi}_{\mathcal{M}}$ is the space of *all* student policies for the original CMDP $\mathcal{M}$ (not only feasible ones), $\mathcal{A}^T = \mathcal{I}$ is the set of all teacher interventions, $\mathcal{P}^T : \overline{\Pi}_{\mathcal{M}} \times \mathcal{I} \times \overline{\Pi}_{\mathcal{M}} \rightarrow [0,1]$ is governed by the student's learning algorithm, $\mathcal{O}^T = \Phi$ is the space of evaluation statistics the teacher gathers, and $\mathcal{Z}^T = \phi$ is the mapping from the student's policies to statistics about them, governed by the teacher's evaluation setup. The reward function $\mathcal{R}^T$ can be defined as $\mathcal{R}^T(n) = \hat{V}(\pi_{n,j}) - \hat{V}(\pi_{n-1,j})$, with $\mathcal{R}^T(0) = \hat{V}(\pi_{0,j})$ the "progress" in student's policy quality from one curriculum stage to the next. Note, however, that what really matters to the teacher is the student's perceived policy quality at the end of round, $\hat{V}(\pi_{N_s,j}) = \sum_{n=1}^{N_s} \mathcal{R}^T(n)$. Thus, in general, a teacher's decision rule is a solution to this POMDP:

**Definition 2** (*Curriculum policy*). *Let* $\mathcal{H}$ *be the space of teacher's observation histories. A curriculum policy is a mapping* $\pi^T : \mathcal{H} \rightarrow \mathcal{I}$ *that, for any* $n \in [N_s]$, *specifies an intervention given the teacher's observation history* $\phi(\pi_1), \ldots, \phi(\pi_{n-1})$ *at the start of the student's* $n$-*th interaction unit.*

In the context of curriculum learning, modeling the teacher as a POMDP agent similar to ours was proposed in [31, 34] – though not for RL safety. However, from the computational standpoint, CISR's view of a teacher as an online learning agent captures a wider range of possibilities for the teacher's practical implementation. For instance, it suggests that it is equally natural to view the teacher as a bandit algorithm that plays suitably parameterized curriculum policies in each round, which is computationally much more tractable than using a full-fledged POMDP solver. As described in Section 4, this is the approach we take in this work.

**Safely evaluating students' policies (ii).** Optimizing the curriculum policy requires the evaluation of students' policies to create features and rewards for the teacher. Since a student's intermediate policy may not be safe w.r.t. $\mathcal{M}$, evaluating it in $\mathcal{M}$ could lead to safety violations. Instead, we assume that the teacher's intervention set $\mathcal{I}$ includes a special intervention $i_0$ satisfying Proposition 2. Hence, this intervention induces a CMDP $\mathcal{M}_0$ where the student can't violate the safety constraint. The teacher uses it for evaluation, although it can also be used for student learning. Therefore, for any policy $\pi$ over the state-action space $\mathcal{S} \times \mathcal{A}$, we define $\hat{V}(\pi) \triangleq V_{\mathcal{M}_0}(\pi)$. Since $\mathcal{M}_0$ has more

constraints than $\mathcal{M}$, evaluation in $\mathcal{M}_0$ underestimates the quality of the teacher's curriculum policy. Let $\hat{\pi}^*$ be the optimal policy for $\mathcal{M}_0$ and $\pi^*$ the optimal policy for $\mathcal{M}$. The value of the teacher's curriculum policy is, then, $\hat{V}(\hat{\pi}^*) \leq V_{\mathcal{M}}(\pi^*)$. If the student policy violates a constraint in $\mathcal{M}_0$ during execution, the teacher gets a reward $-2TR_{\max}$ where $R_{\max}$ is the largest environment reward.

## 4  Implementation Details

CISR allows for many implementation choices. Here, we describe those used in our experiments.

**Student's training and knowledge transfer.** Our students are CMDP solvers based on [30], but train online rather than offline as in [30] since safety is guaranteed by the teacher. This is a primal-dual solver, where the primal consists of an unconstrained RL problem including the original rewards and a Lagrange multiplier penalty for constraint violation. The dual updates the multipliers to increase the penalty for violated constraints. We use the Stable Baselines [25] implementation of PPO [43] to optimize the Lagrangian of a CMDP for a fixed value of the multipliers, and Exponentiated Gradient [27], a no-regret online optimization algorithm, to adapt the multipliers. Our students transfer both value functions and policies across interventions, but reset the state of the optimizer.

**Teacher's observation.** Before every switch to a new intervention $i_{n+1}$, our teacher evaluates the student's policy in CMDP $\mathcal{M}_{i_n}$ induced by the previous intervention. The features estimated in this evaluation, which constitute the teacher's observation $o_n^T$, are $V_{\mathcal{M}_{i_n}}(\pi)$, the student's policy value in CMDP $\mathcal{M}_{i_n}$, and the rate of its constraint violation there, $\mathbb{E}_{\rho_{i_n}^\pi}[\sum_{t=0}^T \mathbb{I}(s_t \in \mathcal{D}_{i_n}) - \tau_{i_n}]$.

**Reward.** As mentioned in Sec. 3.3, from a computational point of view it is convenient to approach the curriculum policy optimization problem as an online optimization problem for a given parametrization of the teacher's curriculum policy. This is the view we adopt in our implementation, where in round $j$, the teacher's objective is the value of the student's final policy $\hat{V}(\pi_{N_s,j})$. Moreover, since after $N_s$ curriculum steps the student's training is over, we compute the student's return directly in $\mathcal{M}$ rather than using a separate evaluation intervention $\mathcal{M}_0$.

**Policy class.** To learn a good teaching policy efficiently, we restrict the teacher's search space to a computationally tractable class of parameterized policies. We consider reactive policies that depend only on the teacher's current observation, $o_n^T$, so $\pi^T(o_0^T, o_1^T, \ldots, o_n^T) = \pi^T(o_n^T)$. Moreover, we restrict the number of times the teacher can switch their interventions to at most $K \leq N_s$. A policy from this class is determined by a sequence of $K$ interventions and by a set of rules that determines when to switch to the next intervention in the sequence. Here, we consider simple rules that require the average return and constraint violation during training to be greater/smaller than a threshold. Formally, we denote the threshold array as $\omega \in \mathbb{R}^2$. The teaching policy we consider switches from the current intervention to the next when $\phi(\pi_{n,j})[0] \geq \omega[0] \wedge \phi(\pi_{n,j})[1] \leq \omega[1]$. Thus, teacher's policies are fully determined by $3K + 1$ parameters.

**Teacher's training with GP-UCB.** Given the teacher's policy is low dimensional and sample efficiency is crucial, we use Bayesian optimization (BO) [32] to optimize it. That is, we view the $3K + 1$ dimensional parameter vector of the teacher as input to the BO algorithm, and search for parameters that approximately optimize the teacher's reward. Concretely, we use GP-UCB [44], a simple Bayesian optimization algorithm that enjoys strong theoretical properties.

## 5  Experiments

We present experiments where CISR efficiently and safely trains deep RL agents in two environments: the *Frozen Lake* and the *Lunar Lander* environments from Open AI Gym [10]. While *Frozen Lake* has simple dynamics, it demonstrates how safety exacerbates the difficult problem of exploration in goal-oriented environments. *Lunar Lander* has more complex dynamics and a continuous state space. We compare students trained with a curriculum optimized by CISR to students trained with trivial or no curricula in terms of safety and sample efficiency. *In addition, we show that curriculum policies can transfer well to students of different architectures and sensing capabilities (Table 1).* **For a detailed overview of the hyperparameters and the environments, see Appendices A and B.**

*Frozen Lake.* In this grid-world environment (Fig. 1a), the student must reach a goal in a 2D map while avoiding dangers. It can move in 4 directions. With probability $80\%$ it moves in the desired direction and with $10\%$ probability it moves in either of the orthogonal ones. The student only sees

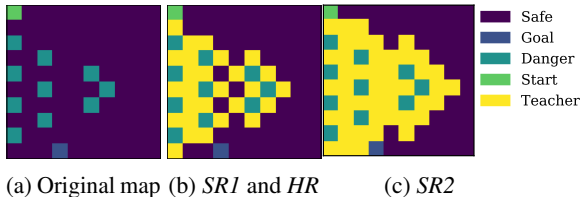
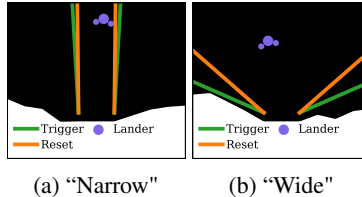

(a) Original map (b) *SR1* and *HR*   (c) *SR2*

Figure 1: Interventions for *Frozen Lake*. Maps 1b and 1c show trigger state sets $\mathcal{D}_{SR1}$ and $\mathcal{D}_{SR2}$ for interventions $SR1$ and $SR2$, which get triggered at distance = 1 and 2 from lakes (dangers), respectively. Intervention $HR$ has $\mathcal{D}_{HR} = \mathcal{D}_{SR1}$.

(a) "Narrow"     (b) "Wide"

Figure 2: Interventions for *Lunar Lander*. If the student hits the green line, it gets reset to a state on the orange line. See Appendix B for more details.

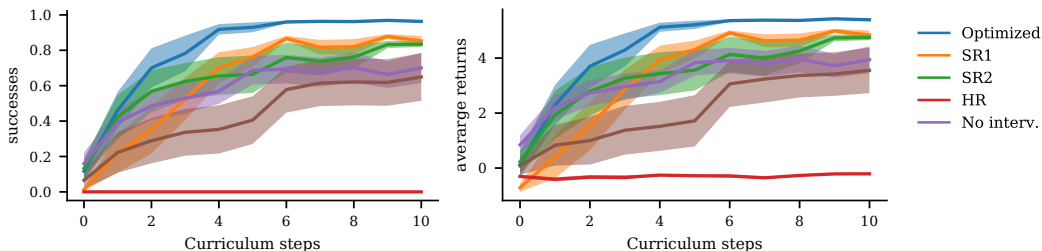

Figure 3: Student success rate (**Left**) and average returns (**Right**) in *Frozen Lake* as student training under different curriculum policies progresses. The *Optimized* curriculum policy outperforms training in the original environment (*No-interv.*), in each of the individual interventions (*SR1*, *SR2*, *HR*) and the pre-specified curriculum policy of [31] (*Bandit*).

.

the map in Fig. 1a and is not aware of the teacher interventions' trigger states (Figs. 1b and 1c). Note that the high density of obstacles, the strong contrast between performance and safety, the safe training requirement and the high-dimensional observations (full map, as opposed to student location) make this environment substantially harder than the standard *Frozen Lake*.

We use three interventions, whose trigger states are shown in Figs. 1b and 1c: *soft reset 1 (SR1)*, *soft reset 2 (SR2)*, and *hard reset (HR)*. SR1 and SR2 have tolerance $\tau = 0.1$ and reset the student to the state where it was the time step before being rescued by the teacher. *HR* has zero tolerance, $\tau = 0$ and resets the student to the initial state.

We compare six different teaching policies: (*i*) *No-intervention*, where students learn in the original environment; (*ii-iii-iv*) *single-intervention*, where students learn under each of the interventions fixed for the entire learning duration; (*v*) *Bandit*, where students follow curricula induced by the *a priori* fixed curriculum policy from [31]; (*vi*) *Optimized*, where we use the curriculum policy found by CISR within the considered policy class after interacting with 30 students. We let each of these curriculum policies train 10 students. For the purposes of analysis, we periodically freeze the students' policies and evaluate them in the original environment.

**Results.** Fig. 3 shows the success rate and the return of the students' policies deployed in the original environment as training progresses. Without the teacher's supervision (*No-interv.*), the students learn sensible policies. However, the training is slow and results in thousands of failures (Table 6 in Appendix B). The *HR* intervention resets the students to the initial state distribution whenever the teacher constraint is violated. However, since it has more trigger states than *No-interv.* (Fig. 1b vs Fig. 1a), the students training exclusively under it do not explore enough to reach the goal. *SR1* and *SR2* allow the student to learn about the goal without incurring failures thanks to their reset distribution, which is more forgiving that *HR*'s one. However, they result in performance plateaus as the soft reset of the training environment lets the students recover from mistakes in a way the deployment environment doesn't. The *Optimized* curriculum retains the best of both worlds by initially proposing a soft reset intervention that allows the agent to reach the goal and subsequently switching to the hard reset such that the training environment is more similar to the original one. Finally, the *Bandit* curriculum policy from [31] requires an initial exploration of different interventions with each student as it does not learn across students. This is in contrast with CISR, which exploits information acquired from previous students to improve its curriculum policy, and results in slower training. Table 6 in Appendix B shows the confidence intervals of mean performance across 10 students and teachers trained on 3 seeds, indicating CISR's robustness.

| | Eval. on noiseless, 2-layer student | | | | Eval. on noisy student | | | | Eval. on 1-layer student | | | |
|---|---|---|---|---|---|---|---|---|---|---|---|---|
| | $V_{\mathcal{M}}(\pi_{N_s})$ | Succ. | Test fail | Train fail | $V_{\mathcal{M}}(\pi_{N_s})$ | Succ. | Test fail | Train fail | $V_{\mathcal{M}}(\pi_{N_s})$ | Succ. | Test fail | Train fail |
| *Optimized* | 233.8 | 88.4% | 10.5% | 1.6 | 221.1 | 86.7% | 10.5% | 2.95 | 254.5 | 92.2% | 6.6% | 2.2 |
| *Narrow* | 183.0 | 72.4% | 30.0% | 1.0 | 149.1 | 65.1% | 32.4% | 1.2 | 220.4 | 83.3% | 15.9% | 1.3 |
| *Wide* | 210.6 | 81.4% | 17.6% | 3.4 | 153.9 | 75.1% | 14.0% | 4.05 | 119.9 | 67.4% | 17.6% | 4.0 |
| *No-interv.* | 236.3 | 90.1% | 7.9% | 1228.8 | 210.3 | 85.5% | 11.8% | 1651.9 | 248.7 | 92.4% | 6.1% | 1368.1 |

Table 1: *Lunar Lander* final performance summary. **Noiseless, 2-layer student (Left):** The *Wide* and *Narrow* interventions result in low performance due to the difficulty of exploration and early plateau, respectively. Students that learn under the *Optimized* curriculum policy achieve a comparable performance to those training under *No-intervention*, but suffer three orders of magnitude fewer training failures. **Noisy student (Center), One layered-student (Right):** The results are similar when we use the curriculum optimized for students with noiseless observations and a 2-layer MLP policy for students with noisy sensors or a 1-layer architecture.

***Lunar Lander***. In this environment, the goal is to safely land a spaceship on the Moon. Crucially, the Moon surface is rugged and differs across episodes. The only constant is a flat landing pad that stretches across $\mathcal{X}_{land}$ along the $x$ dimension (at $y = 0$). Landing safely is particularly challenging since agents do not observe their distance from the ground, only their absolute $x$ and $y$ coordinates.

Since the only way to ensure safe training is to land on the landing pad each time, we use the two interventions in Fig. 2. Each gets triggered based on the student's tilt angle and $y$-velocity if the student is over the landing pad ($x \in \mathcal{X}_{land}$), and on a funnel-shaped function of its $x, y$ coordinates otherwise ($x \notin \mathcal{X}_{land}$). The former case prevents crashing onto the landing pad, the latter, landing (and possibly crashing) on the rugged surface the student cannot sense. We call the interventions *Narrow* and *Wide* (see Fig. 2); both set the student's velocity to 0 after rescuing it to a safe state. The interventions, despite ensuring safety, make exploration harder as they make experiencing a natural ending of an episode difficult.

We compare four curriculum policies: (*i*) *No-intervention*, (*ii-iii*) *single-intervention* and (*iv*) *Optimized*. We let each policy train 10 students and we compare their final performance in the original *Lunar Lander*. Moreover, we use the *Optimized* curriculum to train different students than those it was optimized for, thus showing the transferability of curricula.

**Results.** Table 1 (left) shows for each curriculum policy the mean of the students' final return, success rate and failure rate in the original environment and the average number of failures during training. The *Narrow* intervention makes exploration less challenging but prevents the students from experiencing big portions of the state space. Thus, it results in fast training that plateaus early. The *Wide* intervention makes exploration harder but it is more similar to the original environment. Thus, it results in slow learning. *Optimized* retains the best of both worlds by initially using the *Narrow* intervention to speed up learning and subsequently switching to the *Wide* one. In *No-interv.*, exploration is easier since the teacher's safety considerations do not preclude a natural ending of the episode. Therefore, *No-interv.* attains a comparable performance to *Optimized*. However, the absence of the teacher results in three orders of magnitude more training failures.

Table 1 shows the results of using the teaching policy optimized for students with 2-layer MLP policies and noiseless observations to train students with noisy sensors (center) and different architectures (right). These results are similar to those previously observed: the *Optimized* curriculum attains a comparable performance to the *No interv.* training while greatly improving safety. This shows that teaching policies can be effectively transferred, which is of great interest in many applications. Table 7 in Appendix B shows the confidence intervals for these experiments over 3 seeds.

## 6  Concluding remarks

In this work, we introduce CISR, a novel framework for safe RL that avoids many of the impractical assumptions common in the safe RL literature. In particular, we introduce curricula inspired by human learning for safe training and deployment of RL agents and a principled way to optimize them. Finally, we show how training under such optimized curricula results in performance comparable or superior to training without them, while greatly improving safety.

While it yields promising results, the teaching policy class considered here is quite simplistic. The use of more complex teaching policy classes, which is likely to require a similarity measure between interventions and techniques to reduce the teacher's sample complexity, is a relevant topic for future research. Finally, it would be relevant to study how to combine the safety-centered interventions we propose with other task-generating techniques that are used in the field of curriculum learning for RL.

## Acknowledgments and Disclosure of Funding

We would like to thank Cathy Wu (MIT), Sanmit Narvekar (University of Texas at Austin), Luca Corinzia (ETH Zurich) and Patrick MacAlpine (Microsoft Research) for their comments and suggestions regarding this work. Alekh Agarwal and Andrey Kolobov also thank Geoff Gordon, Romain Laroche and Siddhartha Sen for formative discussions. This work was supported by the Max Planck ETH Center for Learning Systems. This project has received funding from the European Research Council (ERC) under the European Unions Horizon 2020 research and innovation program grant agreement No 815943.

## Broader Impact Statement

Our paper introduces conceptual formulations and algorithms for the safe training of RL agents. RL has the potential to bring significant benefits for society, beyond the existing use cases. However, in many domains it is of paramount importance to develop RL approaches that avoid harmful side-effects during learning and deployment. We believe that our work contributes to this quest, potentially bringing RL closer to high-stakes real-world applications. Of course, any technology – especially one as general as RL – has the potential of misuse, and we refrain from speculating further.

## Footnotes

[2]https://github.com/zuzuba/CISR_NeurIPS20

[3]That is, it finds a feasible policy after enough interactions with the CMDP, assuming one exists.

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
