[Supplementary Material]

# APPENDIX

## A Hyperparameters

In this section, we report the hyperparameters that we use for the students, which are CMDP solvers based on an online version of [30], and for the teachers, which are based on the GP-UCB algorithm for multi-armed bandits [44].

### A.1 Students

The students comprise two components: an unconstrained RL solver and a no-regret online optimizer. The first component is used to solve the unconstrained RL problem that results from optimizing the Lagrangian of a given CMDP for a fixed value of the Lagrange multipliers. For this, we use the Stable Baselines [25] implementation of the Proximal Policy Optimization (PPO) algorithm [43]. The second component is used to adapt the Lagrangian multipliers online. As suggested in [30], we use the Exponentiated Gradient algorithm [27] for this. In the following, we use the hyperaparameters naming convention from Stable Baselines [25] for PPO and from [30] for Exponentiated Gradient.

**Frozen Lake.** In Table 2, we show the hyperparameters used for the students in the *Frozen Lake* experiments, except for those that determine their policy class. In these experiments, the student's policies are parametrized as convolutional neural networks with 2 convolutional layers followed by a fully connected layer. The first convolutional layer has 32 filters of size 3 and stride 1. The second one has 64 filters of size 3 and stride 1. The fully connected layers contains 32 neurons. We use ReLU as activation function.

**Lunar Lander.** In Table 3, we show the hyperparameters used for the students in the *Lunar Lander* experiments, except for those that determine their policy class. In these experiments, the student's policies are parametrized as MLP networks with 2 hidden layers with 20 neurons each. We use ReLU as activation function.

### A.2 Teachers

Our teachers are based on the the GPyOpt [7] implementation of GP-UCB.

The teacher's hyperparameters are those of the Gaussian process (GP) model used by GP-UCB. In all the experiments, we use a GP with radial basis function (RBF) kernel with automatic relevance determination (ARD) and a Gaussian likelihood. Therefore, the teacher has the following hyperparameters: the signal variance, $\sigma_f^2$, an array of $3K + 1$ lengthscales $l \in \mathbb{R}^{3K+1}$, where $3K + 1$ is the number of parameters that determines the teacher's policy for a fixed number of intervention switches, $K$, see Sec. 4, and the noise variance $\sigma_n^2$. Rather than fixing the hyperparameters a priori, we define hyperpriors over them and use their maximum a posteriori (MAP) estimate, which we update after every newly acquired data point.

The data is normalized before being fed to the GP.

**Frozen Lake.** In the *Frozen Lake* experiments, we allow for up to two switches between interventions; that is, $K = 2$. Therefore, $l \in \mathbb{R}^7$. In Table 4, we show the mean and the variance of the Gamma hyperprior of each hyperparameter.

**Lunar Lander.** In the *Lunar Lander* experiments, we allow for up to one switch between interventions; that is, $K = 1$. Therefore, $l \in \mathbb{R}^4$. In Table 5, we show the mean and the variance of the Gamma hyperprior of each hyperparameter.

| Name | Value |
|---|---|
| $n\_steps$ | 128 |
| $ent\_coef$ | 0.05 |
| $learning\_rate$ | 0.001 |
| $noptepochs$ | 9 |

(a) PPO

| Name | Value |
|---|---|
| $B$ | 0.5 |
| $\eta$ | 1.0 |

(b) Exponentiated Gradient

Table 2: Student's hyperparameters for the *Frozen lake* environment.

| Name | Value |
|---|---|
| $n\_steps$ | 500 |
| $ent\_coef$ | 0.001 |
| $learning\_rate$ | 0.005 |
| $noptepochs$ | 32 |

(a) PPO

| Name | Value |
|---|---|
| $B$ | 120 |
| $\eta$ | 1.0 |

(b) Exponentiated Gradient

Table 3: Student's hyperparameters for the *Lunar Lander* environment.

| Hyperparameter | $\sigma_f^2$ | $l_1$ | $l_2$ | $l_3$ | $l_4$ | $l_6$ | $l_5$ | $l_7$ | $\sigma_n^2$ |
|---|---|---|---|---|---|---|---|---|---|
| $\mu$ | 1 | 1 | 0.05 | 1 | 0.05 | 0.2 | 0.2 | 0.2 | 0.01 |
| $\sigma^2$ | 0.2 | 1 | 0.02 | 1 | 0.02 | 0.2 | 0.2 | 0.2 | 0.1 |

Table 4: Mean and variance of the Gamma hyperpriors for the teacher's hyperparameters for the *Frozen Lake* environment.

# B  Experiments

In this section, we provide a detailed explanation of our experimental setup and we present the results we obtained repeating the curriculum optimization and evaluation for multiple random seeds.

## B.1  Frozen Lake

**Environment.** In the *Frozen Lake* experiments, we use the $10 \times 10$ map in Fig. 1a. The student receives the full map as observation but is not aware of the areas of influence of the teacher, Figs. 1b and 1c. In each location, it can take one of four actions: *up*, *right*, *left* or *down*. With probability $80\%$ it moves in the desired direction and with $10\%$ probability it moves in either of the orthogonal ones. After each move, it can end up in one of three kind of tiles: *goal*, which results in a successful termination of the episode, *danger*, which results in a failure and the consequent termination of the episode and *safe*. The agent receives a reward of 6 for reaching the goal and $-0.01$ otherwise (entering dangerous tiles is discouraged via the constraint rather than with low rewards).

An interaction unit between the student and the teacher consists of 10000 time steps. A curriculum lasts for 11 of such interaction units.

**Teacher's training.** We consider curriculum policies that allow for up to two intervention switches, i.e., $K = 2$. To initialize the GP model, we sample 10 curriculum policies at random, train a student with each of those and feed their final performance to the GP model. To optimize the curriculum, we run GP-UCB for 20 iterations, where each iteration corresponds to training a single student with the curriculum policy proposed by GP-UCB.

**Teacher's evaluation.** To evaluate the quality of a curriculum policy, we get 10 new students, we let them train with the curricula induced by such policy and we record their failures during training as well as their returns and their successes when they are deployed in the original environment (i.e. without supervision) for 10000 time steps. Fig. 3 reports the mean of these quantities over the 10 students for all the curriculum policies that we consider at the end of each interaction unit. Notice that evaluating after each intervention unit is done solely for analysis purposes as, in practice, one should not deploy a student in the original environment before the curriculum is completed.

For each of the teaching policies considered, we report the mean returns and success rates at the end of the curriculum over the 10 students as well as their mean number of failures during training in Table 6. Here, the confidence intervals are obtained by optimizing 3 curriculum policies independently with different random seeds and repeating the evaluation procedure for each one.

## B.2  Lunar Lander

The observation space of the *Lunar Lander* environment is 8-dimensional and it includes: $x$ and $y$ position, tilt angle, linear and angular velocities and two Booleans that indicate whether each leg is in contact with the ground. At each time step, the lander can take one of four actions: fire the main, the left or the right engine or do nothing. The agent receives a reward of 100 for a successful landing,

| Hyperparameter | $\sigma_f^2$ | $l_1$ | $l_2$ | $l_3$ | $l_4$ | $\sigma_n^2$ |
|---|---|---|---|---|---|---|
| $\mu$ | 1 | 20 | 1 | 0.2 | 0.2 | 0.01 |
| $\sigma^2$ | 0.2 | 4 | 0.3 | 0.2 | 0.2 | 0.1 |

Table 5: Mean and variance of the Gamma hyperpriors for the teacher's hyperparameters for the *Lunar Lander* environment.

|  | Success | Training failures | $V_{\mathcal{M}}(\pi_{N_s})$ |
|---|---|---|---|
| *Optimized* | **0.960 ± 0.004** | 0 ± 0 | **5.368 ± 0.025** |
| *SR2* | 0.827 ± 0.027 | 0 ± 0 | 4.669 ± 0.168 |
| *SR1* | 0.850 ± 0.011 | 0 ± 0 | 4.839 ± 0.065 |
| *HR* | 0.000 ± 0.000 | 0 ± 0 | −0.222 ± 0.013 |
| *Bandit* | 0.574 ± 0.049 | 0 ± 0 | 3.077 ± 0.288 |
| *No-interv.* | 0.768 ± 0.028 | 3075.6 ± 492.1 | 4.329 ± 0.160 |

Table 6: Final deployment performance in *Frozen Lake* with confidence intervals obtained by training and evaluating the teachers with three different random seeds. The students trained with the optimized curriculum outperform both naive curricula and training in the original environment in terms of success rate and return. All the agents supervised by a teacher are safe during training. In contrast, training directly in the original environment results in many failures. These results are consistent across random seeds, thus showing the robustness of CISR.

of -0.3 for firing the main engine and of -0.03 for firing the side engines. Additionally, there is a potential based reward shaping that encourages the contact of the legs with the ground and moving toward the origin, i.e., the center of the landing pad. At the beginning of each episode, a random force is applied to the agent and the surface of the Moon is generated at random, with the only constant being the flat surface of the landing pad in the center of the map. Since the agent does not observe its distance from the ground and since the surface of the Moon is generated at every episode, the only way to guarantee safety is to land on the landing pad. In the original environment, each episode can terminate with either a successful landing or a failure (either a crash or exiting the game window from the sides, which we call an out of map, OOM, outcome). However, since the teacher's interventions make it hard for inexperienced students to come across a natural ending of the episode, we introduce a timeout, which we set to 500 during training and to 2000 during deployment (a well trained agent usually requires between 150 and 250 steps to land). Every time an episode ends because of a timeout, the student receives a reward of -100.

The trigger function of the interventions depends on whether the student's is above the landing pad, i.e., $x \in \mathcal{X}_{\text{land}}$, or not. In particular, let us denote with $x$ and $y$ the position of the agent, with $\dot{x}$ and $\dot{y}$ its linear velocities, with $\alpha$ its tilt angle and with $\dot{\alpha}$ its angular velocity. The landing pad stretches between $-0.2$ and $0.2$, while the whole map extends from $-1$ to $1$. For a fixed steepness of the funnel $a$, the trigger function of the interventions are of the form:

$$trigger(x, y, \dot{x}, \dot{y}, \alpha, \dot{\alpha}) = \begin{cases} \mathbb{I}(\dot{y} \geq 0.3 + 10y) \vee \mathbb{I}(\alpha \geq 0.5 + 10y) & \text{if } x \in [-0.2, 0.2] \\ y \leq a(-0.2 - x) & \text{if } x < -0.2 \\ y \leq a(x - 0.2) & \text{if } x > -0.2 \end{cases} \quad (3)$$

The reset distribution that determines the student's state after the teacher intervenes also depends on whether the teacher rescues the student above the landing pad or not. We denote with $(x, y, \dot{x}, \dot{y}, \alpha, \dot{\alpha})$ the state where the students gets rescued and with $(x', y', \dot{x}', \dot{y}', \alpha', \dot{\alpha}')$ the state where the student gets reset. First of all, the teacher always stabilizes the student and, therefore, we have $\dot{x}' = \dot{y}' = \alpha' = \dot{\alpha}' = 0$. Thus, the reset distributions only differ based on the location where the teacher steers the student to make it stay clear from danger. In particular, if $x \in [-0.2, 0.2]$, we have $x' = x$ and $y' = y - 0.1$. However, if $x > 0.2$, the reset location of the student is determined by a geometric construction: we reset the student at the intersection between the line of that forms and angle of $135°$ with the horizontal axis passing through $x$ and $y$ and the line $a'(x - 0.2)$, for a given $a' > a$ (the orange line in Fig. 2). A symmetric construction is used in case $x < -0.2$.

The *Narrow* intervention corresponds to $a = 20$ and $a' = 100$, while the *Wide* intervention corresponds to $a = 0.5$ and $a' = 1$.

An interaction unit between the student and the teacher consists of 100000 time steps. A curriculum lasts for 15 of such interaction units.

|  | Succ. | Crash | OOM | $V_{\mathcal{M}}(\pi_{N_s})$ | Training failures |
|---|---|---|---|---|---|
| *Optimized* | $89.2 \pm 0.4\%$ | $9.5 \pm 0.6\%$ | $0.4 \pm 0.1\%$ | $236.1 \pm 0.9$ | $1.7 \pm 0.13$ |
| *Wide* | $79.2 \pm 2.2\%$ | $18.7 \pm 1.5\%$ | $0.4 \pm 0.1\%$ | $199.5 \pm 8.3$ | $3.5 \pm 0.15$ |
| *Narrow* | $72.7 \pm 3.1\%$ | $23.4 \pm 2.1\%$ | $0.4 \pm 0.2\%$ | $187.2 \pm 9.0$ | $0.9 \pm 0.02$ |
| *No-interv.* | $88.5 \pm 1.6\%$ | $7.3 \pm 0.1\%$ | $1.9 \pm 1.4\%$ | $225.1 \pm 7.4$ | $1251.0 \pm 33.72$ |

(a) Evaluation on noiseless, 2-layer students.

|  | Success | Crashes | OOM | $V_{\mathcal{M}}(\pi_{N_s})$ | Training failures |
|---|---|---|---|---|---|
| *Optimized* | $83.4 \pm 2.3\%$ | $13.2 \pm 2.2\%$ | $0.3 \pm 0.1\%$ | $211.5 \pm 6.8$ | $2.6 \pm 0.23$ |
| *Wide* | $78.8 \pm 2.6\%$ | $13.8 \pm 0.8\%$ | $0.7 \pm 0.4\%$ | $184.7 \pm 21.8$ | $4.2 \pm 0.12$ |
| *Narrow* | $63.2 \pm 1.3\%$ | $32.9 \pm 1.0\%$ | $0.6 \pm 0.2\%$ | $139.1 \pm 7.1$ | $1.8 \pm 0.46$ |
| *No-interv.* | $86.0 \pm 0.4\%$ | $10.8 \pm 0.1\%$ | $0.8 \pm 0.2\%$ | $214.7 \pm 3.1$ | $1695.8 \pm 31.04$ |

(b) Evaluation on 2-layer students with noisy sensors.

|  | Success | Crashes | OOM | $V_{\mathcal{M}}(\pi_{N_s})$ | Training failures |
|---|---|---|---|---|---|
| *Optimized* | $92.1 \pm 1.6\%$ | $5.1 \pm 0.6\%$ | $0.0 \pm 0.0\%$ | $253.4 \pm 5.0$ | $1.9 \pm 0.13$ |
| *Wide* | $72.2 \pm 2.9\%$ | $16.6 \pm 0.9\%$ | $2.4 \pm 1.1\%$ | $151.6 \pm 18.5$ | $3.5 \pm 0.25$ |
| *Narrow* | $81.7 \pm 1.0\%$ | $16.3 \pm 0.2\%$ | $0.0 \pm 0.0\%$ | $221.0 \pm 2.1$ | $1.3 \pm 0.03$ |
| *No-interv.* | $94.5 \pm 1.0\%$ | $4.5 \pm 0.9\%$ | $0.1 \pm 0.0\%$ | $256.4 \pm 3.7$ | $1175.0 \pm 80.56$ |

(c) Evaluation on noiseless, 1-layer students.

Table 7: *Lunar Lander* final deployment performance summary for three different kinds of students with confidence intervals obtained by training and evaluating the teachers with three different random seeds. **Noiseless, 2-layer student (Top):** The *Narrow* intervention helps exploration but results in policy performance plateau, the *Wide* one slows down student learning due to making exploration more challenging, and the *Optimized* teacher provides the best of both by switching between *Narrow* and *Wide*. Students that learn under the *Optimized* curriculum policy achieve a comparable performance to those training under *No-intervention*, but suffer three orders of magnitude fewer training failures. **Noisy student (Center), One layered-student (Bottom):** The results are similar when we use the curriculum optimized for students with noiseless observations and a 2-layer MLP policy for students with noisy sensors (center) or a 1-layer architecture (bottom), thus showing teaching policies can be transferred across classes of students. These results are consistent across random seeds, thus showing the robustness of CISR.

**Teacher's training.** We consider curriculum policies that allow for up to one intervention switch, i.e., $K = 1$. Since the student's learning dynamics are quite noisy in this environment, we evaluate each curriculum policy for a class of 10 students in parallel and use the mean final performance of the students as a signal for GP-UCB. To initialize the GP model, we use 4 curriculum policies, one for each possible combination of interventions allowed by the policy class considered. To optimize the curriculum, we run GP-UCB for 10 iterations, where each iteration corresponds to training a class of 10 students in parallel with the curriculum policy proposed by GP-UCB.

**Teacher's evaluation.** The evaluation of teaching policies is analogous to the *Frozen Lake* case: we let each curriculum policy train 10 newly sampled students and we deploy them in the original environment for 200000 time steps to measure their performance. Since a much longer deployment time compared to *Frozen Lake* is required to obtain accurate estimates of the student's performance, we only record it at the end of the curriculum rather than after each interaction unit.

In these experiments, we also investigate the transferability of teaching policies, which is of great importance for many practically relevant scenarios. To this end, we apply the teaching policy optimized for students with perfect state information to students with noisy sensors. In particular, we consider students that observe $\tilde{x} = x + w_x$ and $\tilde{y} = y + w_y$, where $w_x \sim \mathcal{N}(0, 10^{-4})$ and $w_y \sim \mathcal{N}(0, 10^{-4})$. This level of noise is quite challenging as one standard deviation covers 2.5% of the width of the landing pad. In these experiments, the teacher uses the noiseless state information to rescue the student. This captures a scenario that is common in real-world applications where we have hardware that helps preserving safety during training, such as motion capture systems, that we cannot use during deployment. Since training in these conditions is harder and more prone to constraint violation, we let the training run for 20 interaction units rather than 15 and we allow for

higher penalty for constrain violation through the Lagrange multipliers by considering a higher upper bound on them (we set $B = 160$ rather than $B = 120$).

In a separate experiment, we apply the teaching policies optimized for the student's architecture presented in Appendix A to students that only have one hidden layer with 20 neurons rather than two.

For each of the curriculum policies considered and for each of the experiments described above, we report the mean returns, success rates and failure rates at the end of the curriculum over the 10 students as well as their mean number of failures during training in Table 7. Notice that the fact that the rates do not sum to 100% is due to timeouts. The confidence intervals in Tables 7a–7c are obtained by optimizing 3 curriculum policies independently with different random seeds and repeating the evaluation procedure for each one.

## C  Proof

In this section, we provide proofs for Propositions 1 and 2.

**Proposition 1** (*Eventual safety*). *Let $\Pi_{\mathcal{M}}$ and $\Pi_{\mathcal{M}_i}$ be the sets of feasible policies for the problems in Equations* (1) *and* (2), *respectively. Then, if $\tau_i + \kappa_i \leq \kappa$, $\Pi_{\mathcal{M}_i} \subseteq \Pi_{\mathcal{M}}$.*

*Proof.* The main idea of the proof is to show that the constraints in Equation (2), which are based on expectations with respect to $\rho_i^\pi$, are stricter than the constraint in Equation (1), which are based on expectations with respect to $\rho^\pi$. To this end, we need to distinguish between trajectories, or segments thereof, that have the same probability under $\rho^\pi$ and $\rho_i^\pi$ for any $\pi$ from those that do not. Let us denote with $\xi = (s_0, s_1, \ldots, s_T)$, a generic trajectory in $\mathcal{M}$ and with $\Xi$ the set of all possible trajectories in $\mathcal{M}$ (this is the set that the distributions $\rho$ and $\rho_i$ are defined over). Moreover, for a given set of trigger states $\mathcal{D}_i$, we indicate the set of trajectories where at least one state belongs to $\mathcal{D}_i$ with $\Xi_{\mathcal{D}_i} = \{\xi \in \Xi \,|\, \xi \cap \mathcal{D}_i \neq \emptyset\}$ and with $\Xi_{\mathcal{D}_i}^C = \Xi \setminus \Xi_{\mathcal{D}_i}$ its complement. With this notation, the constraint $\mathbb{E}_{\rho_i^\pi} \sum_{t=0}^T \mathbb{I}(s_t \in \mathcal{D}_i) \leq \tau_i$ is equivalent to $\sum_{\xi \in \Xi} \rho_i^\pi(\xi)|\xi \cap \mathcal{D}_i| \leq \tau_i$. Therefore, for a $\pi \in \Pi_{\mathcal{M}_i}$, we know that:

$$\sum_{\xi \in \Xi_{\mathcal{D}_i}} \rho_i^\pi(\xi)|\xi \cap \mathcal{D}_i| + \sum_{\xi \in \Xi_{\mathcal{D}_i}^C} \rho_i^\pi(\xi)|\xi \cap \mathcal{D}_i| \leq \tau_i. \tag{4}$$

Since, by definition, we know that $|\xi \cap \mathcal{D}_i| = 0$ for all $\xi \in \Xi_{\mathcal{D}_i}^C$, (4) simplifies to

$$\sum_{\xi \in \Xi_{\mathcal{D}_i}} \rho_i^\pi(\xi)|\xi \cap \mathcal{D}_i| \leq \tau_i. \tag{5}$$

Every trajectory $\xi \in \Xi_{\mathcal{D}_i}$ can be divided in two segments: $\xi_1 = (s_0, s_1, \ldots, s_m)$, which contains all the states up to the first one in the sequence that belongs to $\mathcal{D}_i$, i.e., $s_0, s_1, \ldots, s_{m-1} \notin \mathcal{D}_i$ and $s_m \in D_i$, and $\xi_2 = (s_{m+1}, \ldots, s_T)$, which contains the remaining part of the trajectory. Thus, we can say:

$$\tau_i \geq \sum_{\xi \in \Xi_{\mathcal{D}_i}} \rho_i^\pi(\xi)|\xi \cap \mathcal{D}_i|, \tag{6}$$

$$= \sum_{(\xi_1,\xi_2) \in \Xi_{\mathcal{D}_i}} \rho_i^\pi(\xi_1, \xi_2)|(\xi_1, \xi_2) \cap \mathcal{D}_i|, \tag{7}$$

$$\geq \sum_{(\xi_1,\xi_2) \in \Xi_{\mathcal{D}_i}} \rho_i^\pi(\xi_1, \xi_2)|\xi_1 \cap \mathcal{D}_i|, \tag{8}$$

$$= \sum_{\xi_1 \in \Xi_{\mathcal{D}_i}} \rho_i^\pi(\xi_1)|\xi_1 \cap \mathcal{D}_i| \sum_{\xi_2 \in \Xi_{\mathcal{D}_i}} \rho_i^\pi(\xi_2|\xi_1), \tag{9}$$

$$= \sum_{\xi_1 \in \Xi_{\mathcal{D}_i}} \rho^\pi(\xi_1)|\xi_1 \cap \mathcal{D}_i| \sum_{\xi_2 \in \Xi_{\mathcal{D}_i}} \rho^\pi(\xi_2|\xi_1), \tag{10}$$

$$= \sum_{(\xi_1,\xi_2) \in \Xi_{\mathcal{D}_i}} \rho^\pi(\xi_1, \xi_2)|\xi_1 \cap \mathcal{D}_i|, \tag{11}$$

$$\geq \sum_{(\xi_1,\xi_2) \in \Xi_{\mathcal{D}_i}} \rho^\pi(\xi_1, \xi_2)|(\xi_1, \xi_2) \cap \mathcal{D}| = \sum_{\xi \in \Xi_{\mathcal{D}_i}} \rho^\pi(\xi)|\xi \cap \mathcal{D}|. \tag{12}$$

In the previous chain of inequalities, (10) holds because $\rho$ and $\rho_i$ are the same for the portion of the trajectory before the teacher intervenes for the first time, i.e., $\xi_1$, and because $\sum_{\xi_2 \in \Xi_{\mathcal{D}_i}} \rho_i^\pi(\xi_2|\xi_1) = \sum_{\xi_2 \in \Xi_{\mathcal{D}_i}} \rho^\pi(\xi_2|\xi_1) = 1$. Furthermore, (11) holds because $|\xi_1 \cap \mathcal{D}_i| = 1$ by definition of $\xi_1$ and because $|(\xi_1, \xi_2) \cap \mathcal{D}| \leq 1$ since the states in $\mathcal{D}$ are terminal.

Moreover, for a $\pi \in \Pi_{\mathcal{M}_i}$, we know that:

$$\sum_{\xi \in \Xi_{\mathcal{D}_i}} \rho_i^\pi(\xi)|\xi \cap \mathcal{D}| + \sum_{\xi \in \Xi_{\mathcal{D}_i}^C} \rho_i^\pi(\xi)|\xi \cap \mathcal{D}| \leq \kappa_i. \tag{13}$$

Since the teacher does not modify the original dynamics unless the student enters a trigger state, we know that $\rho_i^\pi = \rho$ for every $\xi \in \Xi_{\mathcal{D}_i}^C$. Therefore, (13) becomes:

$$\sum_{\xi \in \Xi_{\mathcal{D}_i}} \rho_i^\pi(\xi)|\xi \cap \mathcal{D}| + \sum_{\xi \in \Xi_{\mathcal{D}_i}^C} \rho^\pi(\xi)|\xi \cap \mathcal{D}| \leq \kappa_i. \tag{14}$$

By summing (12) and (14), we obtain:

$$\sum_{\xi \in \Xi_{\mathcal{D}_i}} \rho^\pi(\xi)|\xi \cap \mathcal{D}| + \sum_{\xi \in \Xi_{\mathcal{D}_i}^C} \rho^\pi(\xi)|\xi \cap \mathcal{D}| \leq \tau_i + \kappa_i - \sum_{\xi \in \Xi_{\mathcal{D}_i}} \rho_i^\pi(\xi)|\xi \cap \mathcal{D}| \leq \tau_i + \kappa_i, \tag{15}$$

which means $\mathbb{E}_{\rho^\pi} \sum_{t=0}^{T} \mathbb{I}(s_t \in \mathcal{D}) \leq \tau_i + \kappa_i \leq \kappa$, which implies that $\pi \in \Pi_{\mathcal{M}}$. $\qquad \square$

**Proposition 2 (*Learning safety*).** *Let $\mathcal{D}$ be the set of unsafe states of CMDPs $\mathcal{M}$ and $\mathcal{M}_i$, and let $\mathcal{D}_i$ be the set of trigger states of intervention $i$. If $\mathcal{D} \subseteq \mathcal{D}_i$ and $\mathcal{P}(s'|a, s) = 0$ for every $s' \in \mathcal{D}$, $s \in \mathcal{S} \setminus \mathcal{D}_i$, and $a \in \mathcal{A}$, then an optimal student learning in CMDP $\mathcal{M}_i$ will not violate any of $\mathcal{M}$'s constraints throughout learning.*

*Proof.* During learning, the student may use any policy $\pi$ from the set of all possible policies for the origianl environment $\mathcal{M}$, $\overline{\Pi}_{\mathcal{M}}$ (which includes unfeasible and, therefore, unsafe policies). However, during training, for any intervention $i$ the student transitions according to the dynamics $\mathcal{P}_i$ rather than $\mathcal{P}$. Therefore, we aim to show that $\mathbb{E}_{\rho_i^\pi} \sum_{t=0}^{T} \mathbb{I}(s_t \in D) \leq \kappa$ for all $\pi \in \overline{\Pi}_{\mathcal{M}}$. If $s_t \in \mathcal{S} \setminus \mathcal{D}$, we can either have (*i*) $s_t \in \mathcal{S} \setminus \mathcal{D}_i$ or (*ii*) $s_t \in \mathcal{D}_i \setminus \mathcal{D}$. Let us consider these two cases separately. In (*i*), we know that $\mathcal{P}(s_{t+1}|s_t, a_t) = 0$ for any action $a_t$ and any $s_{t+1} \in \mathcal{D}$ by assumption. Moreover, since $\mathcal{P}(s_{t+1}|s_t, a_t) = \mathcal{P}_i(s_{t+1}|s_t, a_t)$ for all $s_t \notin \mathcal{D}_i$, we have $\mathcal{P}_i(s_{t+1}|s_t, a_t) = 0$ for all $s_t \in \mathcal{S} \setminus \mathcal{D}_i$ and $s_{t+1} \in \mathcal{D}$. In case (*ii*), we know that $\mathcal{P}_i(s_{t+1}|s_t, a_t) = 0$ for all $s_{t+1} \in \mathcal{D}_i \supseteq \mathcal{D}$ and $s_t \in \mathcal{D}_i$ by definition of reset distribution. Therefore, we have shown that for all $s_t \in \mathcal{S} \setminus \mathcal{D}$, $a_t \in \mathcal{A}$ and $s_{t+1} \in \mathcal{D}$, we have $\mathcal{P}_i(s_{t+1}|s_t, a_t) = 0$. As a consequence, the only way the student can reach an unsafe state under the dynamics $\mathcal{P}_i$ is if $s_0 \in \mathcal{D}$, which corresponds to starting an episode in an unsafe terminal state, which only depends on the initial state distribution and not on the policy. If the initial state distribution is such that it is not possible for the student to start an episode in an unsafe state, then we have $\mathbb{E}_{\rho_i^\pi} \sum_{t=0}^{T} \mathbb{I}(s_t \in D) = 0 \leq \kappa$ for every $\pi \in \overline{\Pi}_{\mathcal{M}}$. Otherwise, we have assumed $\kappa$ to be such that the problem in (1) is feasible. Therefore, it must be such that all the trajectories starting with $s_0 \in \mathcal{D}$ can be tolerated. $\qquad \square$