[Reviews · NeurIPS 2020]

Review 1

Summary and Contributions: This paper focuses on the problem of safe RL, with the goal of developing a framework for online agent training where the agent only acts safely both during and after training. It develops the CISR framework, which assumes the existence of an automated "teacher" that can intervene while the agent is learning to reset its state. The teacher has a curriculum policy for how to sequence the interventions. The teacher is also an online learner that optimizes the curriculum policy as it interacts with students. The paper develops the details of this framework, shows that it guarantees safety in training, and in two experiments, show that optimizing the curriculum for the interventions outperforms choosing a single intervention and the final policy is comparably effective to an agent trained without interventions (while maintaining safety).

Strengths: The strengths of this work center on how the authors lay out a clear, general framework for safe RL with somewhat simpler assumptions than in much of the prior literature. The framework is relatively general, with the ability to use different algorithms to fill in for the teacher and student, and the interventions are likely to be realizable in many real applications where safe RL is desired. While the experiments are brief, they nicely illustrate how the framework can be instantiated in practice. These strengths are likely to make the work both of interest to the NeurIPS community and to have an impact on practitioners using these ideas in practice, especially given that while some elements of the framework have the potential to be computationally intense, less computationally intense elements can be used in the implementation (e.g., Bayesian optimization rather than a POMDP solver).

Weaknesses: There were two primary weaknesses that I noticed in the paper: (1) The paper notes that the framework is different from prior curriculum learning work due to learning from prior learners, allowing it to be "data-driven rather than heuristic," but the consequences of that aren't explored. In particular, this may mean that many agents must be trained prior to getting to a good curricular policy, and this may be problematic for practitioners. This fact is somewhat hidden in the experimental evaluations because the evaluation of the optimized sequence of interventions doesn't include the performance of the first 30-100 learners. It would be helpful to either include a clearer argument for the importance of learning the curriculum policy over other approaches or to discuss the possible limitations of needing to learn from many learners (and perhaps the robustness claims would mitigate this limitation to some extent). I would also have liked to see results (perhaps in the appendix) for the sensitivity of the results to the number of learners prior to reaching an "optimized" point. (2) In the experimental results, small K (K = 1 and K = 2) are used. It seems possible that a heuristic would perform very well here. I would have been interested in looking at larger K (or scenarios where larger K are needed) and comparing to a baseline of, say, switching interventions at a uniform interval through the training, to understand how much the experiments are actually telling us about how well the optimization works. Understanding this would also help with understanding whether the time spent to optimize on previous learners was worth it. An additional, perhaps stronger comparison would be to a curriculum learner like [21], [27] that is working on the same student. Update after reading author response: Thank you to the authors for their response. I appreciate especially the inclusion of a comparison to [27], which I believe strengthens the empirical result. The point about the number of possible curriculum sequences makes sense, but I do not believe it necessarily addresses the point about a heuristic performing relatively well. Figure (b) in the response helps to address that there is learning going on across students, but doesn't address whether there's a large proportion of the possible curricula that perform well and thus whether the learning task is relatively easy.

Correctness: To the extent I could tell, the paper+supplement are correct as written. However, I am somewhat confused by the interaction between the paper and the supplement, as the supplement notes that Proposition 1 isn't quite right and says that this will be in the final version of the paper.

Clarity: The paper was clearly written, especially in the description of the CISR framework (up through the end of section 3). The experimental details were brief, although generally well-explained in the appendix. I had two concerns about the exposition: (1) The explanation of the special intervention i_0 and its used in evaluating the policy is somewhat hard to follow when first introduced. (2) For the lunar lander, intuition that the "narrow" intervention is better for exploration was unclear to me. It seemed like more exploration would be possible in the "wider" intervention scenario (although it might not be as well targeted), since if I understand correctly, narrow prevents exploration outside of the very narrow funnel. Clarifying this intuition would be helpful.

Relation to Prior Work: The paper generally made the connections to prior work clear, and covered connections from a variety of different subareas that were relevant to the work.

Reproducibility: Yes

Additional Feedback: Overall, I thought this paper was interesting, easy to follow, and laid out very nice ideas for safe RL. My main feedback is above, and the parts that to me seem like they would be most likely to increase the impact would be to show that this works in a setting where larger K is helpful and/or a larger set of interventions is possibly (basically, showing how well it can scale). In the intro, the possible application to human students is mentioned. While this isn't a main focus, the proposed example seems slightly far afield in terms of "the safety notion consist[ing] in ensuring that the material is presented to a (human) student in an appropriate manner to avoid them dropping the course" - it's not clear, at least to me, how the type of interventions proposed in the paper would be translated to an ITS for human learners. It would be nice to temper this example a bit. For the broader impact statement, although this is not factored into my score, I would have appreciated a more fleshed out statement about possible misuses than "Of course, any technology – especially one as general as RL – has the potential of misuse, and we refrain from speculating further." In particular, the "we refrain from speculating further" is somewhat off-putting to the reader about whether negative impacts have really been considered. Typos: - Line 314: extra period after GP-UCB. - Supplementary material, lines 534 and 602: "different random sees"


Review 2

Summary and Contributions: This paper describes an approach to curriculum learning based on the idea of a supervisor algorithm. They identify a low-dimensional class of supervisor sequences (i.e., a parameterization of curricula) and then optimize that with an GP-UCB. They show that this approach can improve learning in Frozen Lake and Lunar Lander.

Strengths: The paper is working on a problem of interest for the NeurIPS community and I believe the suggestion is novel. In principle, this type of approach could be quite useful in the long run.

Weaknesses: While this is a nice idea, I think the theoretical grounding and empirical evaluation are not up to the NeurIPS standards. I like that the authors chose to frame the curriculum learning problem as a POMDP, but the description had clarity issues and wasn't grounded in an example. The authors use a bandit algorithm to optimize this POMDP, but don't discuss the potential drawbacks of using this approach heuristically (or show bounds on performance). The experiments provided are a good proof-of-concept, but I think that, absent strong theoretical results, they are not enough to validate the proposed method. A convincing implementation of this method in more complex domains (e.g., autonomous driving) would help a lot. Finally, the authors need to do a better job of motivating their design choices and assumptions. This is particularly at issues in l.181 (Assumption 2), where a fairly impactful assumption about supervisor policies is made without discussing the feasibility or applicability of this assumption. Similarly, the assumption to frame this problem in a situation with multiple students is not really motivated. It is clear why this is useful practically, but the authors need to describe a better real world use-case.

Correctness: The claims and method appear to be correct, although some of the design choices are under motivated. The empirical methodology is weak but correct.

Clarity: The paper has a lot of room for clarity improvements. I advise the authors to include an overview figure to introduce and go over the proposed approach and to consider a running example to explain the main ideas. The paper would also benefit from restructuring in order to emphasize technical details of the method and the experiments. This means condensing the first half of the paper substantially and expanding the second half (at a minimum, including a discussion of experimental results). The paper also needs work to have more signposts to help readers. For example, the teachers objective is not described until l.221. That description is incomplete and describes the objective as evaluating the performance by computing features.

Relation to Prior Work: The related work is reasonably complete. It gives a good overview, but does not do a good job of comparing this work to prior work. I think one notable exception is work from the human factors engineering community, which looks at safety during learning [1]. [1] Akametalu, Anayo K., et al. "Reachability-based safe learning with Gaussian processes." 53rd IEEE Conference on Decision and Control. IEEE, 2014.

Reproducibility: Yes

Additional Feedback: After the discussion with other reviewers, it is clear that I had some misunderstandings about the intended application of this work. I have updated my review to be aligned with the consensus of the other reviews.


Review 3

Summary and Contributions: The author response addressed most of my concerns. I agree that an evaluation with a larger number of possible interventions would be beneficial. I think the overall idea is nice and while there are some issues with presentation and clarity, I think these can be easily fixed. --- This paper addresses the general problem of safe reinforcement learning, where there are a set of safety constraints that should not be violated by the agent. While most safe RL methods either make mistakes during training or require strong priors, the proposed method uses a teacher to help the agent stay safe during learning. The teacher learns to select from a set of interventions to keep the agent safe during training and helping the agent learn a safe policy that can be executed at test time when the teacher is not present.

Strengths: This work addresses an important problem in RL. Allowing agents to learn safely via stabilizing controllers while enabling test time safety without any stabilizing controllers is an important contribution and has applications to real world RL systems such as robotics.

Weaknesses: While I like the overall idea, I feel that the experiments involve simply curriculum policies with just a few parameters and seem too simple to justify the broad and general claims made in the introduction. The assumption that the teacher has an intervention set that enables them to always save the learning agent seems like a strong assumption in practice. In the experiments it seems really hard to hand engineer these safety controllers to save the agent, unless the designer has intimate knowledge of the task and how to nicely guide the agent to a good solution. Given the assumptions on the teacher in lines 209--216, I am not convinced that the method's proposed in the paper will work. The authors first claim that the teacher does not know the dynamics, but without knowledge of the dynamics how a teacher or intervention designer hope to find the set D_i that blankets D? The authors also assume that the teacher does not know the reward function of the learning agent, but the proposed methods depend on knowing the value function of the agent to measure progress.

Correctness: \hat{V} seems impossible to get if the teacher doesn't know the learning agent's reward function. Line 214: The authors say here that the teacher cannot violate the dynamics of the CMDP. However, many of the interventions involve teleporting the agent to a safe place and in other parts of the text the authors say the teacher can override the dynamics. As written, I don't think Proposition 1 is correct. D_i has not been rigorously defined yet, so we don't know that D \subseteq D_i like we do in Prop 2. Without this assumption we may have an agent that doesn't violate D_i but does violate D. --- Thank you for your detailed comments. I encourage the authors to consider rephrasing things for better clarity. In particular the section "what does a teacher not have to know" says the teacher doesn't need to know the student's reward. While this is true, the experimental results rely on the teacher knowing the expected return of the agent so this statement does not match the rest of the paper.

Clarity: The paper is well written and mostly easy to follow.

Relation to Prior Work: This paper could do a better job at comparing and contrasting with recent work on shielding for RL, learning a curriculum, safety constraint satisfaction during RL, and machine teaching: Alshiekh et al. "Safe Reinforcement Learning via Shielding" AAAI, 2018. -Formal methods for interventions to make RL training safe. Sanmit Narvekar and Peter Stone. "Learning Curriculum Policies for Reinforcement Learning." AAMAS, 2019 -They also pose curriculum learning as an RL problem for the curriculum designer. Thananjeyan et al. "Safety Augmented Value Estimation from Demonstrations (SAVED): Safe Deep Model-Based RL for Sparse Cost Robotic Tasks." RAL, 2019. -Uses model-based RL to avoid safety constraints while learning. Brown and Niekum, "Machine Teaching for Inverse Reinforcement Learning: Algorithms and Applications" -Considers how a teacher can best teach a sequential decision maker.

Reproducibility: Yes

Additional Feedback: What if some states are unsafe but not terminal? How does this relate to option learning? It seems like the interventions are just options. Line 261: "decision" It would be nice for explainability to see the final optimized policy for the teacher to gain insights into the learned curriculum.


Review 4

Summary and Contributions: This paper presents an approach for ensuring safety in RL algorithms, where two distinct agents are defined, an agent who learns a traditional RL policy and an agent who supervises this agent and learns how to teach efficiently. This proposed approach ensures that safety constraints are met, while optimizing the curriculum policy with respect to the student's overall performance. The contributions of the paper are demonstrated in experiments in a simulated environment. The authors addressed some of my concerns in the author response. I would still find it interesting if they discussed that the first few students would have worse experiences than the latter students and if this might have ethical consequence in a real life setting.

Strengths: The strengths of the work are to introduce a novel algorithm for learning policies in a safe environment and jointly learning how to optimize the instruction of these policies such that the student stays safe but learns efficiently. This framework has medium significance. A potential strength is in decoupling the learning of the teacher and the learning of the student. The teacher does not need to know the students reward or transition functions. This can be a strength as a simpler teacher model can be more efficient to train and program. However it isnt' clear if anything is lost by this formulation. The role of the teacher means that students can learn safely online which is appealing.

Weaknesses: The authors do not compare to other state of the art methods in the field. The evaluation section could be improved if they did. For example they could compare to Le et al. [26]. The policy class of the teacher could be explained more fully. How would the threshold array be defined? Is it input to the algorithm or a parameter? It seems like the features that the teacher uses to change interventions have nothing to do with the environment itself. Is this correct? If so why not have features of the environment so that the teacher can learn functions to describe which kinds of states are unsafe? Although a strength of the algorithm is that it can safely learn in online settings all of the results are in simulated settings. It is a weakness that no experiments are shown in real-world online settings.

Correctness: To the best of my knowledge the methodology is correct.

Clarity: Overall the paper is well written. The key ideas could be made a little more clear. For example, in the key ideas section an example of performance could be given. What does it mean for the student to perform well in the bicycle example and do the authors assume that performance is generally in contradiction to safety? Typo: "Here, we describe the those used in our experiments." I would like to see the use of interventions written a little more clearly. What exactly is counted as a student using a teacher's help? Every time they transition according to \mathcal{T}, or every time a new CMDP is defined?

Relation to Prior Work: Yes, the major distinction from prior work is to train the teacher online. However, empirical comparisons to prior work would improve the paper.

Reproducibility: Yes

Additional Feedback:

[Author Response · NeurIPS 2020]



(a) Overview figure

(b) Teacher learning curve for *Frozen lake*: the student return induced by the teaching policy at the end of the curriculum improves as CISR trains more students.

(c) Comparison to TS curriculum in [27] (Bandit). For CISR, we evaluate a teacher policy trained w/30 students on new *test students*, while Bandit learns by explore-exploit for each student as [27] can't learn from previous students. This results in weak performance, esp. with HR proposed at the start of training resulting in poor student policies.

**Common.** Thank you for your helpful comments!

*Synthetic experiments (R2, R4).* As you noted, the main contribution of this work is conceptual. As such, our
experiments are in line with other conceptual CL-for-RL and safe-RL papers (e.g., [2,6,9,16,26,33)]), which evaluate
on the same or similar problems due to their illustrative benefits. While synthetic, they are difficult especially from the
safety standpoint. We agree it would be great to apply CISR to, say, an autonomous driving setting, but doing so would
require motivating so many application-specific engineering choices that it is best done in a separate paper (e.g., [23]).

*Using multiple students (R1, R2).* Using multiple students enables CISR's key novelty – allowing the teacher to *learn*
a curriculum policy in a data-driven way. In contrast, in single-student CL such as [21,27] the teacher continually
estimates the student's partially observed internal state and *heuristically applies* interventions based on these estimates,
but the state estimates $\rightarrow$ interventions mapping – the curriculum policy – is fixed, encoded into the teacher's algorithm.
Among other things, this lets CISR produce curriculum policies that are robust to student diversity (see Table 1 caption,
last 3 lines). *This makes CISR applicable,e.g., in a flavor of sim-to-real transfer where a curriculum policy is learned in*
*a crude simulator and then deployed for training real-world agents in safety-sensitive settings such as robotics.*

*Empirical benefits of multiple students and comparison to prior work (R1, R2, R4).* The reviews gave great ideas for
improving these aspects, and we ran additional experiments, to be included in the revised paper version. Fig. (c) shows
a comparison to [27] and Fig. (b) shows how the teacher improves with multiple students.

*Proposition 2 and Assumption 2 (R2, R3).* Prop 2 only says that *if* interventions are absolutely safe *then* CISR ensures
absolute training safety. Assumption 2 is for conceptual simplicity, but can hold in reality: systems such as aircraft stall
prevention and collision avoidance guarantee near-absolute safety. Even in the absence thereof, CISR, informally, keeps
the student as safe during training as teacher's interventions allow. In Lunar Lander experiments, intervention safety is
not absolute ("Train fail" column in Table 1), but CISR still improves training safety by 1000x over existing approaches.

*Clarity (R1, R2, R4).* We'll add signposts, including Fig. (a) above, examples, and rework the interventions explanation.

**R1:** While simpler heuristics might be possible for small $K$, note that, e.g., in Frozen Lakes the curriculum space is
large even for $K = 2$ and just 3 interventions. With 10 curriculum steps per student, there are $9 \cdot 10/2 = 45$ choices for
2 switching points, each with $3! = 6$ intervention orderings. Thus, we have at least 270 possible curricula. The fact that
CISR determines a good one after only 10 students attests to its learning ability.

**R2:** Please see the **common** responses above. We'll add the related work you mentioned.

**R3:** *Teacher's dynamics knowledge.* The teacher doesn't need it because it just applies pre-designed controllers in
pre-specified states. Designing such controllers may require knowing *local* dynamics around dangerous states, but this
is still far less restrictive than a full dynamics knowledge assumption.

*"Good" behavior of controllers.* We also don't assume the interventions to be "good" in the sense of reward performance.
E.g,. in reality, emergency breaking may induce undesirable behavior. Their main role is just to keep the student safe,
so that it can eventually learn to avoid triggering these interventions in the first place.

*Teacher can't violate dynamics.* What we mean is that the reset distribution $\mathcal{T}(\cdot, s)$ should be realizable based on the
CMDP's dynamics and the teacher's primitive actions, e.g., by using an option. E.g., a parent helping a child stay
upright when riding a bicycle doesn't violate physics, just applies actions unavailable to the child.

*Teacher's reward knowledge.* We only assume that the teacher has *a* reward notion – which may not match the student's
– in order to guide the learning process.

*Proposition 1* is correct because the constraint over $\mathcal{D}$ is also present in the intervention CMDP in Eq (2).

**R4:** *The threshold array* is part of the teaching policy parameters and it's learned via Bayesian optimization, see Fig (b).

*Interventions:* Under intervention $i$, the student triggers the teacher's help whenever it enters $\mathcal{D}_i$ and transitions
according to $\mathcal{T}_i$. The features for the teacher depend on the constraint violation (l. 297) and, thus, on what is dangerous
in the environment according to the interventions (learning the interventions is not our focus but it has been done [15]).

*We'll incorporate remaining reviewer comments, including related work, into the paper as well.*

[Meta-Review · NeurIPS 2020]

After the discussion all reviewers support acceptance, noting that the paper lays out a novel, clear, and general framework for safe online RL. This topic is very relevant to the NeurIPS community, and the paper should be disseminated. However, all reviewers expressed at least minor concerns. I strongly encourage the authors to consider this feedback so that they can improve the responses from future readers. In the discussion, it also became clear that a reviewer thought the paper described an agent interacting with *human* students - perhaps future clarifications can avoid this point of confusion.